# Neurofuzzy Data Aggregation in a Multisensory System for Self-Driving Car Steering

**Antonio Luna-Álvarez** [1] , **Dante Mújica-Vargas** [1,*] , **Arturo Rendón-Castro** [1] , **Manuel Matuz-Cruz** [2] **and Jean Marie Vianney Kinani** [3]

[1] Department of Computer Science, Tecnológico Nacional de México/CENIDET, Interior Internado Palmira S/N, Palmira, Cuernavaca 62490, Mexico

[2] Departamento de Sistemas Computacionales, Tecnológico Nacional de México/ITTapachula, Tapachula Chiapas 30700, Mexico

[3] Unidad Profesional Interdisciplinaria de Ingeniería Campus Hidalgo, Instituto Politécnico Nacional, Pachuca 07738, Mexico

\* Correspondence: dante.mv@cenidet.tecnm.mx

**Abstract:** In the self-driving vehicles domain, steering control is a process that transforms information obtained from sensors into commands that steer the vehicle on the road and avoid obstacles. Although a greater number of sensors improves perception and increases control precision, it also increases the computational cost and the number of processes. To reduce the cost and allow data fusion and vehicle control as a single process, this research proposes a data fusion approach by formulating a neurofuzzy aggregation deep learning layer; this approach integrates aggregation using fuzzy measures $\mu$ as fuzzy synaptic weights, hidden state using the Choquet fuzzy integral, and a fuzzy backpropagation algorithm, creating a data processing from different sources. In addition, implementing a previous approach, a self-driving neural model is proposed based on the aggregation of a steering control model and another for obstacle detection. This was tested in an ROS simulation environment and in a scale prototype. Experimentation showed that the proposed approach generates an average autonomy of 95% and improves driving smoothness by 9% compared to other state-of-the-art methods.

**Keywords:** self-driving; deep learning; neurofuzzy data processing; fuzzy backpropagation; self-driving simulation; scale prototype

## 1. Introduction

In systems, the use of multiple data sources allows an actuator to have a broad view of the environment it is seeking to control, thus allowing it to perform this task with greater precision and to tolerate anomalies in data from some sources [1]. This method is used in physical and robotic systems to widen the field of perception and avoid blind spots, while the redundancy of the data allows the controller system to detect anomalies and filter them [2]. Mobile robots use this method to perceive their environment from different sensors that provide spatial, inertial, and visual information [3]; this is most notable in the Autonomous Vehicles application. The research area for vehicles control is divided into sub areas such as: environment perception, object and vehicles recognition, behaviors, planning and route selection, lane maintenance, signal detection, steering and speed control, among others [4]. This research focuses on the sensor data fusion for steering control.

Various strategies have been proposed in the literature to control the self-driving vehicle direction, just to name a few: starting with the fuzzy inference systems [5,6], which controls the direction by means of inference rules. In simulations, the predictive models [7,8] are recurrently used to trace a route to follow within the path, as well as its combination with fuzzy logic [9]. The aforementioned methods share the characteristic of requiring knowledge transfer from the algorithm designer, other similar strategies are path detection [10], knowledge transfer classifiers [11], and semantic segmentation [12]. On the other hand,

self-tuning methods are the most widely used, from strategies based on reinforcement learning [13–16], which are generally executed in simulations since they require learning based on trial and error or mistakes. Another type of reinforcement learning is based on vision [17,18], although there are other methods based on pure vision [19,20]. The aforementioned strategies mainly focus on keeping the vehicle on track; however, the current trend for steering control is mainly focused on deep learning. This deep neural domain can be divided into control strategies through vision and detection with convolutional networks [21–23]. Other alternatives within the same paradigm are recurrent networks for a visual–temporal relationship [24,25]. The previously mentioned works of the deep learning paradigm present good results; however, they use a single data source such as camera vision. In the literature, it is possible to notice that the methods that present the best results are those based on multisensory systems, for example those that combine visual and spatial information [26–28], the a priori data fusion methods by object detection [29], and fusion 3D [30]. Returning to the deep learning paradigm, another approach to process multiple data sources is to create parallel architectures that process each source separately and integrate the outputs [31–33], as these are the ones with the highest computational cost. To reduce computational cost, deep learning [34,35] fusion methods are used, although they depend on a second process to perform post-fusion control.

Although the aforementioned works present good results, they have some weaknesses. In parallel models, the computational cost increases according to the increase in sensors to be processed, making this option viable only in simulation, since it is too much load for a system on board the vehicle or embedded at scale. Similarly, a priori fusion methods represent a double computational cost since they must fit the data sources as well as filter and extract features that will be processed by the control algorithm. A controller-integrated fusion method lowers the computational cost, particularly deep learning-based fusion models performed using a neural layer. Ideally they are formulated from the adjustment of synaptic weights $W$, the hidden states $h$, activation functions $\sigma(\cdot)$, and the backpropagation algorithm $\Delta W(\nabla L)$. However, in Ref. [34] it depends on a PID controller coupled to the neural network to perform the direction and speed adjustment, stopping the network process and processing the data fusion separately from the neural model. In Ref. [35] the multiview aggregation approach uses perspective transformation to project $n$ feature maps according to the corresponding $1-n$ camera settings. The N projected feature maps are concatenated and a convolution is used to obtain the result. Like these examples, many others [36–38] perform control using a methodology composed of parallel or sequential processing blocks that interact with the neural model, creating a different dependency on the methods, thus increasing response time, fusion quality, and computational cost.

As an alternative, this article proposes a layer with a neurofuzzy aggregation approach, expressed as a neuronal model layer. The main contributions are the following: (a) the neurofuzzy aggregation layer allows extracting and relating features from different structural shapes sources; (b) it has generalization capacity, so it is viable for data fusion in any application through model training; (c) it allows for multiple dimensions inputs unlike the known neural models; (d) it can be added to an existing model as one more layer; and (e) the implementation is compatible with the TensorFlow framework, so it can be parallelizable on the GPU and is not processed as an external program. This proposal was evaluated in a self-driving vehicle steering control task; the experimentation was carried out in a simulated environment in the Robotic Operating System (ROS), as well as in a scale prototype in a controlled environment.

The rest of the document is organized as follows. Section 2 details the proposed layer formulation, as well as the neural architecture for self-driving. Section 3 details the experiments performed, as well as their validation and comparative analysis with other current methods in the literature. Finally, in Section 4 a synthesis of results is made concluding with a brief discussion and future work.

## 2. Materials and Methods

The main contribution is the data aggregation neural layer formulation, so to understand the proposal it is necessary to know the basic operation of the neural paradigm. This performs only data aggregation; for a more complex task such as self-driving, a more complex model composed of several layers is required. For this reason, this section details the basic concepts, the proposal formulation, and its adaptation for self-driving.

### 2.1. General Neural Layer Formulation

In the ANNs paradigm, a model is understood as a set of algorithms grouped into layers that perform a specific task. A neural layer consists of three main elements: for the synaptic weights, given a $X$ data set structured in tensors of size $n$, there will be a weight $W$ for each $X$ such that $\forall x \in X, \exists w \in W : |X| = |W| \wedge x \neq \varnothing$. The hidden state $h$ obtained by a two Euclidean magnitudes $\mathcal{N}(\cdot, \cdot)$ product function, is defined as a linear algebraic product operation of order $n$ according to the dimensionality of $X$ and $W$. The parameters of such a function are $\mathcal{N}(X, W)$ which generate a continuous $h$ output. Furthermore, the activation function $\sigma(h)$ is understood as a linear, non-linear, binary, or probabilistic function that scales the $h$ state. The output $y$ can be discrete or continuous, depending on $\sigma(\cdot)$ domain and the purpose of the layer application. Depending on the layer application, it is defined as a feature extraction layer, memory, recurring, etc.; however, most of them depend on these 3 essential components. For the layer components to correctly process the data, it is necessary to train the parameters using the backpropagation algorithm. To adjust the weights $W$ it is necessary to know the empirical error generated by the propagation in the training stage, based on the expected outputs $y'$ known as ground truth. The error can be calculated by a distance measure known as loss function $L(W)$, which can be:

$$L(W) = \sum_{i=1}^{I} ||\sigma(h_i) - y'_i||_2^2 \tag{1}$$

although depending on the application it can be considered an absolute, polynomial, or radial Euclidean measure. The general method for minimize the error is to iteratively update the parameters by adding an increment $\Delta W$ to the current value: $W := W + \Delta W$. Conversely, if a function $\mathcal{N}(x, W)$ is used to approximate the output values and it is differentiable with respect to $W$, it is possible to use the Gradient Descending method as a learning algorithm:

$$\Delta W = -\alpha \frac{\partial E(W)}{\partial W} \tag{2}$$

where $0 < \alpha < 1$ is a parameter known as the learning rate, which regulates the updating $\Delta W$ in the error gradient $\nabla$. In this way the general algorithm for the neural layers is described; graphically it is represented by the scheme of Figure 1.

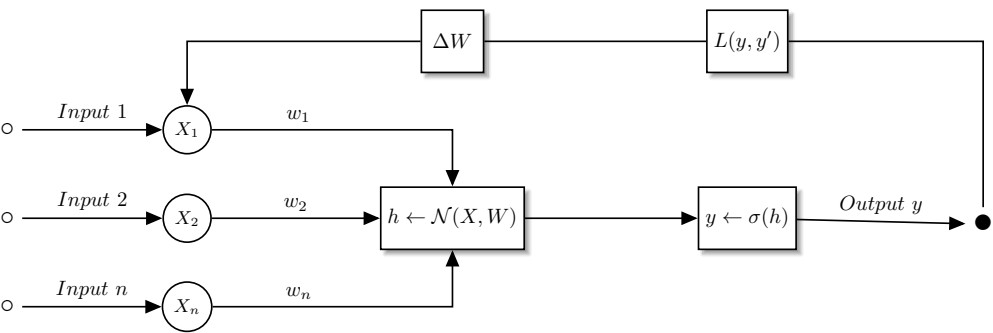

**Figure 1.** General neural layer formalization.

From this base, it is possible to reformulate the algorithm to give a different objective to the classification or characteristics extraction. To perform the aggregation, the neural layer was transformed to the fuzzy domain and its basic components were reformulated to be compatible with fuzzy aggregation measures $\mu$, in addition to working with variable dimensionality inputs. This is described below.

### 2.2. Neurofuzzy Layer Formulation

To summarize the algorithmic adaptations required to formulate the neurofuzzy layer, the proposal components are listed below. Inputs $X$: whether it is a tensor of order 2 or higher, an input of different order is allowed for each element of the first dimension, that is, the signal $x_{1,}$ is structurally different from the one $x_{2,}$, allowing us to operate 2D and 1D signals in the same tensor. The inputs $X$ are structured in an asymmetric tensor $x_s$ of maximum dimensionality $x_s \in \mathbb{R}^{d \times (m \times n)}$, according to the highest order tensor.

Fuzzy weights $W_\mu$: from a fuzzy measure $\mu$ defined as a transformation function within the fuzzy domain $\mu(x) : [0,1]^\eta \mapsto [0,1]$, and interpreted as a weighting function such that $\mu(\emptyset) = 0$, $\mu(X) = 1$, the fuzzy weights $W_\mu$ are defined for each $x \in X$ such that $\sum_{i=0}^{|X|} w_{\mu_i} = 1$. At the same time, there are non-fuzzy weights $W_s$ for each ordered entry $x_s$ and that are adjusted by them, independent of the fuzzy aggregation. Hidden state $h$: depends on the implemented operator, for the neuron generality an algebraic function of the Euclidean magnitudes product can be used. Therefore, the Choquet fuzzy integral defined as:

$$\int f \circ \mu = \sum_{i=1}^{\eta} [f(x_{s(i)}) - f(x_{s(i-1)})] \cdot \mu(x_{s(i)}) \tag{3}$$

where $f(\cdot)$ indicates the ordered data fuzzy transformation and $\mu(\cdot)$ the aggregation using the fuzzy measures $\mu$. Given that $x \in X$ is a tensor of order greater than 0, a reduction is necessary using an operator such as the inner product $\mathcal{N}(\cdot, \cdot)$. This in relation to the introduction of non-fuzzy weights $W_s$ for each $x_s$. Thus, the state $h$ is detailed as the extension of the Choquet integral:

$$h = \sum_{i=1}^{\eta} [\mathcal{N}(x_{s(i)}, W_{s(i)}) - \mathcal{N}(x_{s(i-1)}, W_{s(i-1)})] \cdot \mu(X, W_\mu) \tag{4}$$

in such a way that the fuzzy integral $\mu(\cdot, \cdot)$ depends on the linear product $\mathcal{N}(\cdot, \cdot)$, which in the case must be an outer product $\times$ due to the asymmetry of the input tensors. Activation $\sigma(h)$: As this neuron is not dedicated to classification, a probabilistic function such as Softmax is not used, so a non-linear or rectifying function must be used like any hidden layer. Training algorithm: the propagation of the error within a complete model is carried out in the normal way except for this layer, for which it is necessary to update both the $W_s$ and the fuzzy $W_\mu$ weights. Regarding Momentum, its fuzzy version resides in the scaling of the gradient $\nabla$, this is modified by the scalar value of the fuzzy integral obtained from the aggregation product:

$$m_{\mu_{i+1}} = \alpha m_{\mu_i} + \eta \nabla \mu(x_s, w_s) \tag{5}$$

where $\nabla \in [0,1]$ and the diffuse momentum $m_\mu \in [-1,1]$ are restricted. In this way the adjustment $\Delta W_\mu$ is given by:

$$\Delta W_\mu = W_\mu - \alpha(y \cdot m_\mu) - \alpha(1-y))\nabla \mu(X, W_\mu). \tag{6}$$

On the other hand, the loss function must remain in the fuzzy domain, which is why the MSE can generate alterations to the adjustment by being able to obtain values greater than 1. For this reason, the Logarithm of the Hyperbolic Cosine is used:

$$L(W) = \sum_{i=0}^{n} \log\left(\frac{e^{(y'-y)} + e^{-(y'-y)}}{2}\right) \tag{7}$$

In summary, in the Algorithm 1 the propagation and backpropagation algorithm of this proposal is presented.

---

**Algorithm 1** Neurofuzzy $\Delta W_\mu$ training

---

**Require:** $e \in \mathbb{Z}, \alpha \in \mathbb{R}, X_1, X_2, X_n \in \mathbb{R}^+$
**Ensure:** $y$
1:   $W_s \leftarrow random([-1,1])$
2:   $W_\mu \leftarrow 0$
3:   $x_s \leftarrow X_1 \times X_2 \times X_n : m \times N = \{n \in \mathbb{N}| \ |N| = m\}$
4:   **while** converge **do**
5:      **for** $i \leftarrow 0$ **to** $i = n$ **do**
6:         $h_{s_i} \leftarrow \mathcal{N}(w_s, x_{s_i})$
7:         $h_{\mu_i} \leftarrow \mu(x_{s_i}, w_\mu)$
8:         $h_i = \sum_{i=1}^{\eta}[h_{s_i} - h_{s_i - 1}] \cdot h_\mu$
9:         $y_e \leftarrow \sigma(h_i)$
10:      **end for**
11:      $L(W)_e = \sum_{i=0}^{n} \log\left(\dfrac{e^{(y' - \sigma(h\mu))} + e^{-(y' - \sigma(h\mu))}}{2}\right)$
12:      $m_{\mu_{i+1}} = \alpha m_{\mu_i} + \eta \nabla \mu(x_s, w_s)$
13:      $\Delta W_\mu = W_\mu - \alpha(\sigma(h) \cdot m_\mu) - \alpha(1 - \sigma(h))) \nabla \mu(X, W_\mu)$
14:      $\Delta W_s = W_s - \alpha(y_e \cdot m_j) - \alpha(1 - y_e) \nabla L(W)_e$
15:      $e \leftarrow e + 1$
16: **end while**
      **return** $y$

---

In the Algorithm 1 in step 3, the asymmetric tensor structuring is used. This transformation generates a single structure that respects each data source shape, grouping in an additional $m$ dimension. The algorithm requires two product calculations, one linear $h_s$ and one based on fuzzy measure $h_\mu$, later they are unified by means of the fuzzy integral in step 8 in the output structure $h$. It should be noted that an output $y$ is not generated from this algorithm since this belongs to the classification part used in the neural model to be implemented; however, it is necessary for the adjustment of the weights $W_s$. Therefore the neurofuzzy model is defined in Figure 2.

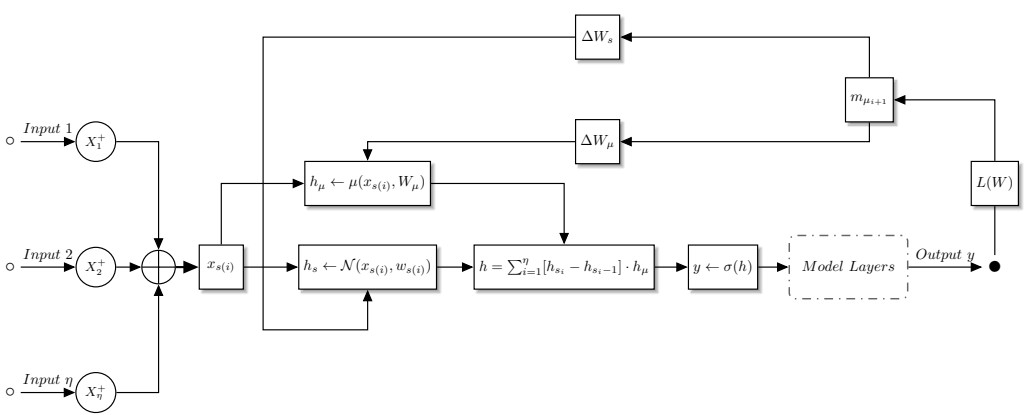

**Figure 2.** Neurofuzzy aggregation layer scheme.

In the above description, the hidden state function $h$ is capable of performing the aggregation of two signals, using only an activation function $\sigma$ rectifier. However, for an application such as self-driving, the model requires greater complexity as well as a greater number of layers with different activations. For this reason, Figure 2 represents the layers of the model in a dotted block, implying that after the aggregation layer there are dense layers, convolutions, or any other item that performs different tasks with information from unified

sources. The resources used to adapt the self-driving model with neurofuzzy aggregation are detailed below.

### 2.3. Self-Driving Model

The proposed fuzzy aggregation layer purpose is to combine multiple signals to reduce the data within a neural model. Thus, it is necessary to have a driving model to guarantee performance and subsequently reduce it. For this, the time-distributed Chauffeur model shown in Figure 3 is used.

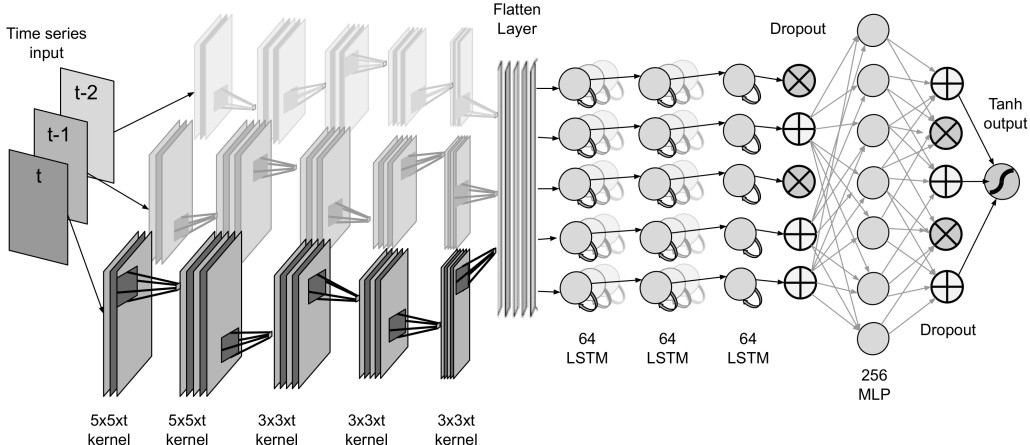

**Figure 3.** Time-distributed Chauffeur model [39].

This model receives information in a sequence of images form in the YUV color space, so it is structured as a 4D tensor of size $t \times 3 \times 200 \times 320$, where $t$ represents the frames per instance, 3 are the color channels, and $200 \times 320$ is the size of each image. The model output is reduced to a scalar obtained from the output layer activation function $\sigma$, given by a Hyperbolic Tangent function that generates outputs in the interval $[-1, 1]$. These values represent degrees of turn in the direction of the vehicle. However, the td-Chauffeur model cannot detect objects. To carry out the object detection from visual information, the Mobilenet model was used as a base, as shown in Figure 4.

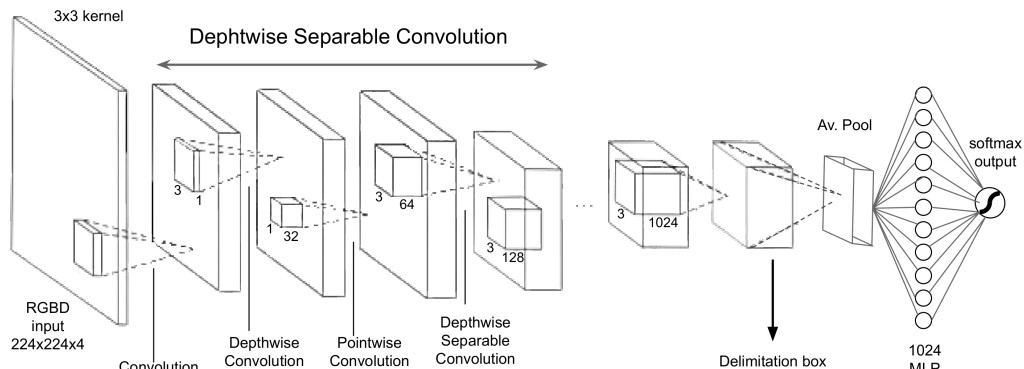

**Figure 4.** Mobilenet model for object detection [40].

This mobile model uses depth separable convolutions. Significantly reduces the number of parameters compared to networks with regular convolutions and same depth, resulting in a lightweight deep neural network. The model basis is to use depth convolution and point convolution layers, so the model can be adapted to the information to be processed. Originally it receives a $224 \times 224 \times 3$ 3D tensor that represents an RGB image; however, for this application it has been adapted to process a $224 \times 224 \times 4$ 3D tensor that

is interpreted as an RGBD image. As an output it obtains the bounding boxes and distance from the center of the objects.

In a simple methodology, it is possible to perform the task by running both models in different threads and processing both outputs in a third algorithm. The limitation of that is the computational cost, in a computer with processing capacity it is possible to run using most of the resources; however, in an embedded system this is limited by: (1) the GPU and CPU processing capacity is with less capacity than a desktop computer, (2) the increased heating and power consumption due to the overuse of resources make the implementation unfeasible, and (3) the models synchronization is a point to consider even on a workstation. For this reason, the integration of both models through an asymmetric tensor distribution is proposed, generating a single multifunction model with layers of neurofuzzy aggregation to reduce the complexity of the model. This proposal can be seen visually in Figure 5.

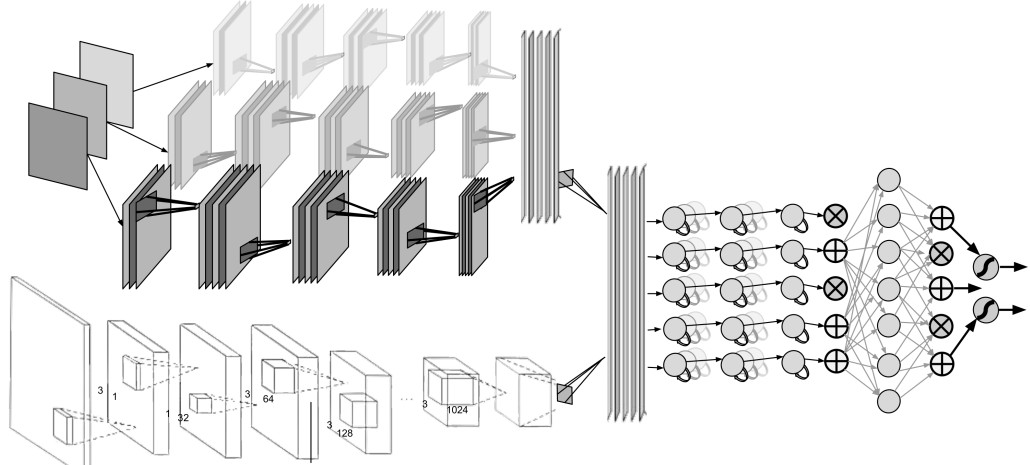

**Figure 5.** Self-driving operational model.

The graphical representation makes it appear that the model is a parallel structure of two neural networks. At the tensor level it is interpreted as two layers of the outermost dimension $n$; however, in the dimension $n-1$ the tensor shape changes in such a way that the input is defined as $I = \{\{224 \times 224 \times 3 \times t\}, \{224 \times 224 \times 4\}\}$. The asymmetry of the input of each neural layer is notorious; however, the reduction in processing threads is carried out by distributing the GPU process only. Where model shrinking is observed is in the $A_{W_\mu}$ layer where the bounding boxes are obtained, the temporal shrinking of the *Flatten* layer and added along with the translation and rotation $I_p = \{\{x^t, y^t, z^t\}, \{x^r, y^r, z^r\}\}$. Finally, the output is extended to two TanH functions to control direction and acceleration.

## 3. Experimental Results

To validate the proposal performance, two experiments were carried out: a simulation in ROS with variables from a real urban environment and a scale prototype tested in a controlled environment. In both cases, scenarios with obstacles and free paths are considered, as well as the use of more than one sensor. To quantify performance, supervised and unsupervised metrics were used, which specialized in measuring the quality, precision, and autonomy of self-driving. Both experiments were evaluated by the following metrics.

### 3.1. Metrics

First, the supervised evaluation was carried out to find out the driving model similarity with the actions carried out by a human driver. For this a ground truth $y'$ was created from capturing steering commands during manual driving on the test path. From this reference it was possible to measure the difference in distance terms between what is obtained and what is expected, for this reason the Mean Square Error is used as evaluation metric, in the same way the Cosine Distance is used:

$$\cos(\theta) = \frac{\sum_{i=0}^{n} y_i \cdot y_i'}{\sqrt{\sum_{i=0}^{n} y_i^2} \cdot \sqrt{\sum_{i=0}^{n} y_i'^2}} \tag{8}$$

the output range spans $[-1, 1]$, where 0 indicates zero error; therefore, 1 and $-1$ indicate full left and right error. The MSE provides an error rate between both types of driving, regardless of the variations produced by time and the speed of movement of the vehicle. For this reason the Driving Behavior metric is used:

$$DB = \left\| \sqrt{\frac{\sum_{i=1}^{n} (y_i - \bar{y})}{n}} - \sqrt{\frac{\sum_{i=1}^{n} (y_i' - \bar{y}')}{n}} \right\| \tag{9}$$

this metric evaluates the entire route by calculating the deviation between both lines; thus, time does not take part in the evaluation and thus providing a speed-invariant distance index. In an unsupervised way, driving is measured by Path Smoothness, i.e., this refers to the angles amplitude that are described while the vehicle is moving:

$$\kappa = \frac{1}{n} \sum_{i=2}^{n} \left[ \arctan\left( \frac{y_{i+1} - y_i}{x_{i+1} - x_i} \right) - \arctan\left( \frac{y_{i-1} - y_i}{x_{i-1} - x_i} \right) \right] \tag{10}$$

where $x_i$ and $y_i$ represent the position of the vehicle in a specific trajectory segment. The metric obtains the angle between two consecutive segments of the path. A lower smoothness value indicates a smoother path. On the other hand, to evaluate the ability to operate independently of a driver, use is made of the autonomy metric proposed by [41]:

$$autonomy = \left( 1 - \frac{interventions \cdot 6}{elapsed\ time} \right) \cdot 100 \tag{11}$$

as well as the Absolute Autonomy Time metric [42], which measure the interventions of a human driver as an error in the system, which is the autonomy time the metric oriented to absolute intervention:

$$AAT = \left( 1 - \frac{tiempo\ de\ intervención}{tiempo\ transcurrido} \right). \tag{12}$$

To complement the experimentation, some methods known from the literature for self-driving were evaluated under the same conditions. Specifically they were the Pilotnet model [41], Donkeycar self driving library [43], and Matlab Automated Driving Toolbox. For these, self-driving models were taken and replicated in the ROS system using it as an intermediary.

### 3.2. Experiment 1: ROS Self-Driving Simulation

This task was carried out based on the free access ROS design, CAT Vehicle Testbed, which consists of a Ford Escape vehicle with the actual physical measurements. The package includes sensors built into the vehicle; however, they were modified to suit this application. Originally featuring a Velodyne LiDAR sensor with a 2000 point cloud, this was modified to produce a 12,000 point cloud with a horizontal aperture of 90° and a vertical aperture of 33.67°. In the same way, the package has two cameras positioned on the vehicle at angles of $-45°$ and 45° with origin in front of the vehicle, these capture RGB images of size $800 \times 800$ pixels. For this application, a single camera pointing to the origin with a size of $640 \times 320$ pixels was required. To evaluate the vehicle autonomy it was necessary to integrate city environments with different characteristics such as intersections, houses and buildings, closed roads, and obstacles in the way. In order to have a variety of possible scenarios, the Vehicle and City Simulation was integrated, which fully represents a small city. Some fragments of the road with obstacles were taken to carry out the self-driving evaluation. These are shown in Figure 6.

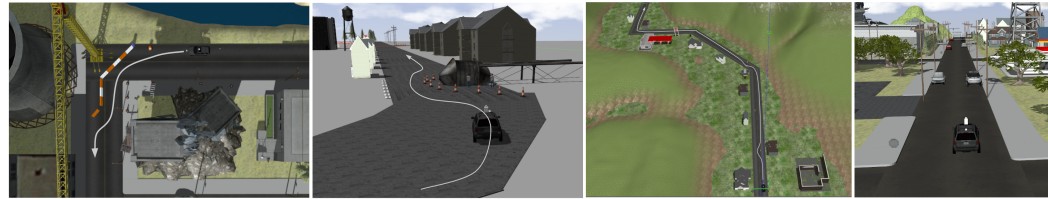

**Figure 6.** ROS simulated environments.

Figure 7 shows the sketches with the shapes and measurements of the simulation segments in which the tests were performed. To validate the effectiveness of each method, two short scenarios and two more complex ones were used, all with objects partially obstructing the path and one with incorrect paths. For all the scenarios there was a trajectory to reach the goal.

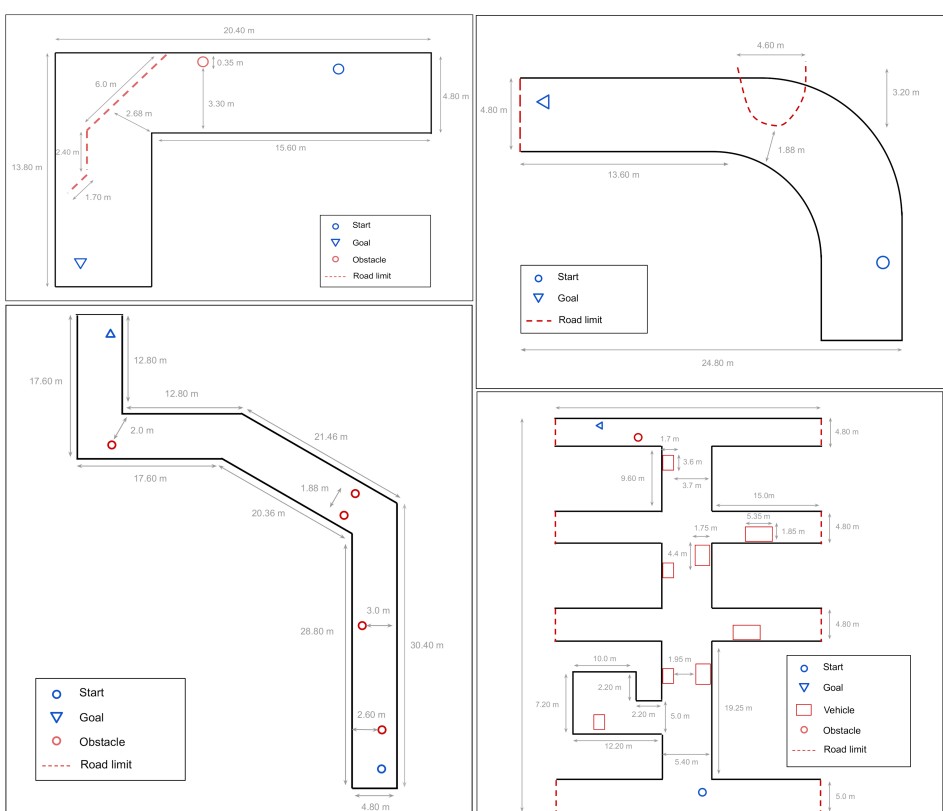

**Figure 7.** Real-scale simulated environments maps for self-driving tests.

Figure 8 shows the conduction trajectories generated by each evaluated method. It can be seen that in the simplest scenarios all the methods can reach the goal; however, it should be noted that the proposal performs the smoothest movements, approaching what is performed by a human, as shown in the ground truth. In the most complex environment it can be observed that a method takes the wrong path, ending the evaluation for it. For the cases in which the algorithm cannot complete the route, it was necessary to enable manual driving to resume the path, an action necessary to calculate the Autonomy and AAT metrics. This intervention is also carried out intentionally when the displacement moves away from the ground truth, thus obtaining an evaluation of autonomy. Quantitatively, the results are summarized in Table 1.

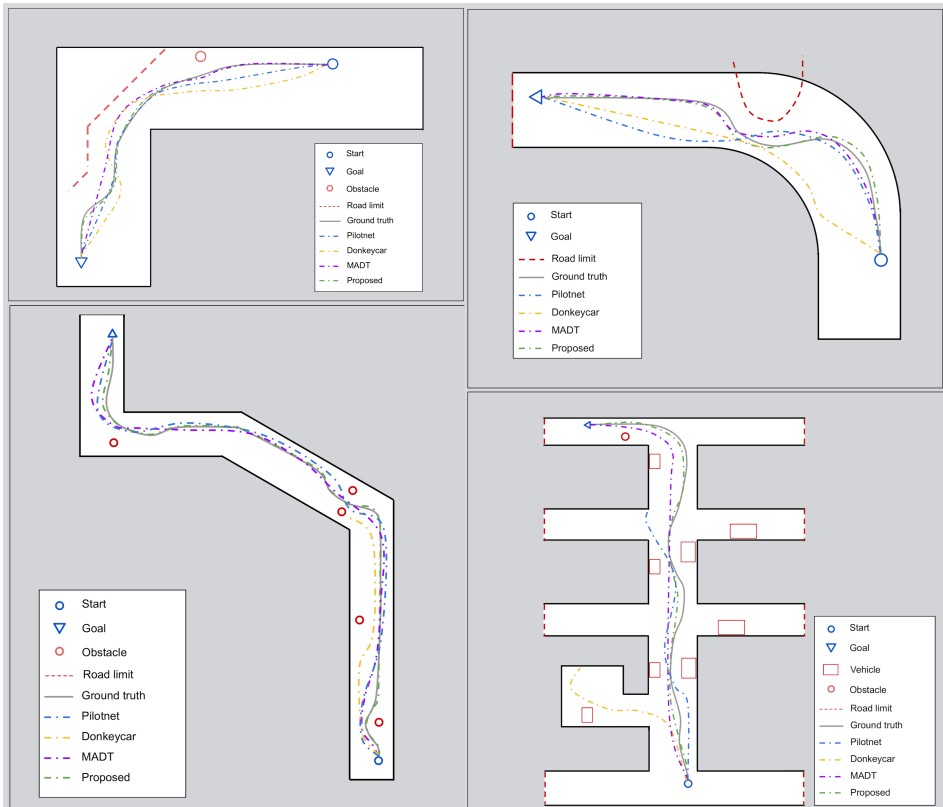

**Figure 8.** Trajectories obtained during self-driving in simulated environments.

According to what is shown in Table 1 and with respect to Figure 8 trajectories, it is observed that the proposal shows the best autonomy metrics are obtained by the proposed method. A range security of 96% is guaranteed, while the worst performance offers only 71%, as long as timely interventions are made, otherwise a collision would result. In simulation, an autonomy of 96.6% is obtained, which indicates that it depends on human intervention for only 3.4% of route, compared to the second best method, which is 8.6% more dependent. In road smoothness, the proposal obtains a metric of 0.256, which is 0.72 lower than the mean of the other methods. This metric indicates that the proposed approach offers a ride that is up to 2.8 times smoother and more comfortable, without any hard rocking.

**Table 1.** Quantitative results of simulation experimentation.

| Method | MSE | Cosine Distance | Behavior | Path Smoothness | Autonomy | AAT |
|---|---|---|---|---|---|---|
| Pilotnet | 0.106 | 0.221, −0.076 | 0.030 | 0.682 | 86.4% | 92.9% |
| Donkeycar | 0.385 | 0.380, −0.510 | 0.236 | 1.826 | 71.3% | 79.8% |
| MADT | 0.077 | 0.112, −0.014 | 0.024 | 0.442 | 88.0% | 94.7% |
| Proposed | 0.021 | 0.011, −0.044 | 0.003 | 0.256 | 96.6% | 97.4% |

To complement the experiments, each conductions method was compared with respect to a human conduction. The metrics showed that the proposed method obtains a mean error of 0.021, where the lowest showed difference of 0.056 compared to the second best. This error indicates that the method provides up to 3.6 times better than human-like driving than another method in this comparison. This measurement is seconded by the driving behavior metric, in which the minimum error of 0.003 is obtained, which indicates that the speed is similar to that of a human driving, without affecting the steering commands smoothness. Finally, the cosine distance showed an imbalance in the lateral displacement error, which was −0.044 and 0.011, so it presented more errors in left turns. This is due to the notable imbalance in the turns of the tracks that can be seen on the maps shown. Still,

it showed the best balance of ≈1–3, compared to MADT which was ≈1–9. Although the Donkeycar method had a relationship ≈1–1.3, the error was up to 10× higher.

As is known, in simulation environments the results are not affected by environmental effects such as lighting, ground textures, obstructions, etc. For this reason, to complement the experimentation, a small scale prototype was designed that integrates in an embedded model light version to control the vehicle direction.

### 3.3. Experiment 2: Scale Prototype Steering Control

In the first instance, the base vehicle was acquired, which includes the mechanics and motors in operation. Its dimensions and appearance shown in Figure 9 represent a 1:16 scale of a compact vehicle in length and width.

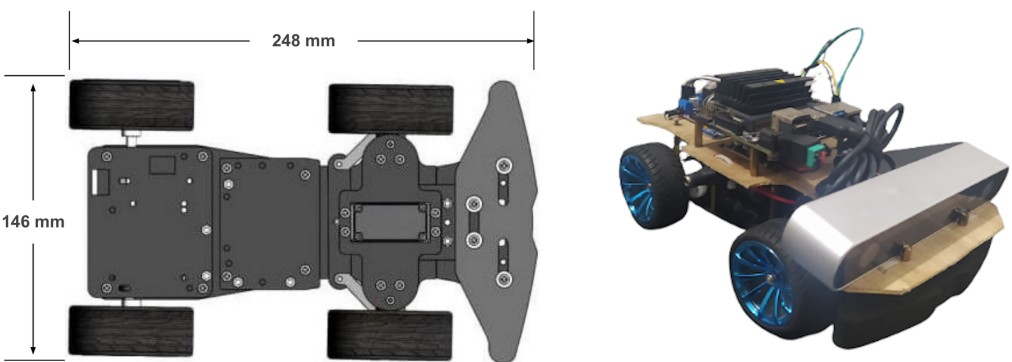

**Figure 9.** Scale prototype designed for self-driving physical tests.

Inspired by the Jetracer prototype, a Jetson Nano Dev Kit 4G embedded card was used as the control module due to its versatility in terms of high-level and low-level programming, as well as its GPU-embedded computing capability on CUDA under Ubuntu. This card includes a 4-core @ 1.43 GHz ARMv7 processor, 4 GB of 1600 MHz RAM, 16 GB eMMC 5.1, GPIO with L2C and PWM support for geared and servo motors, and Nvidia Maxwell 128 CUDA core GPU. In conjunction with the Jetson Nano card, some electronic components were used in the periphery: PCA9685 module, connected to the GPIO ports dedicated to the L2C type connection of the Jetson Nano. This device is used to control the motors by transforming the digital signals to PWM signals that give precision to the servo motor angular speed and position. The device can work at 3.3 V with direct power from the Jetson Nano; however, an external 5 V power is required to move the servo motor. The L298N module serves to magnify the voltage at the 3.3 V input from the PCA9685 to 12 V output. It is used to power the geared motor that pushes the vehicle in a channel.The Lm2596 dc-dc voltage regulator is used to convert 12 V input voltage to 5 V to power the PCA9685 module and the Jetson Nano. Due to the prototype size and power limitations, it used the ZED camera since it allows for obtaining visual information as well as spatial information from the generation of point clouds, in a similar way to the LiDAR sensor. In this way, two types of signals are obtained from the same sensor.

The embedded system works in a similar way to the simulation; however, the processing is conducted on the card, so it was required to interpret the model in the TensorFlow Lite version. Since the information is only obtained from the ZED camera, an acquisition system was designed in Python using the camera drivers. This program acquires the point clouds $I_d = \{M \cdot N \cdot D\}$ and creates the image sequences in tensors of $T$ times at $I_t = \{M \times N \times C \times T\}$. Rotation and translation information is also acquired from the camera controllers. This generates as input three tensors: 4D, 1D, and 2D. Figure 10 shows the implemented system scheme.

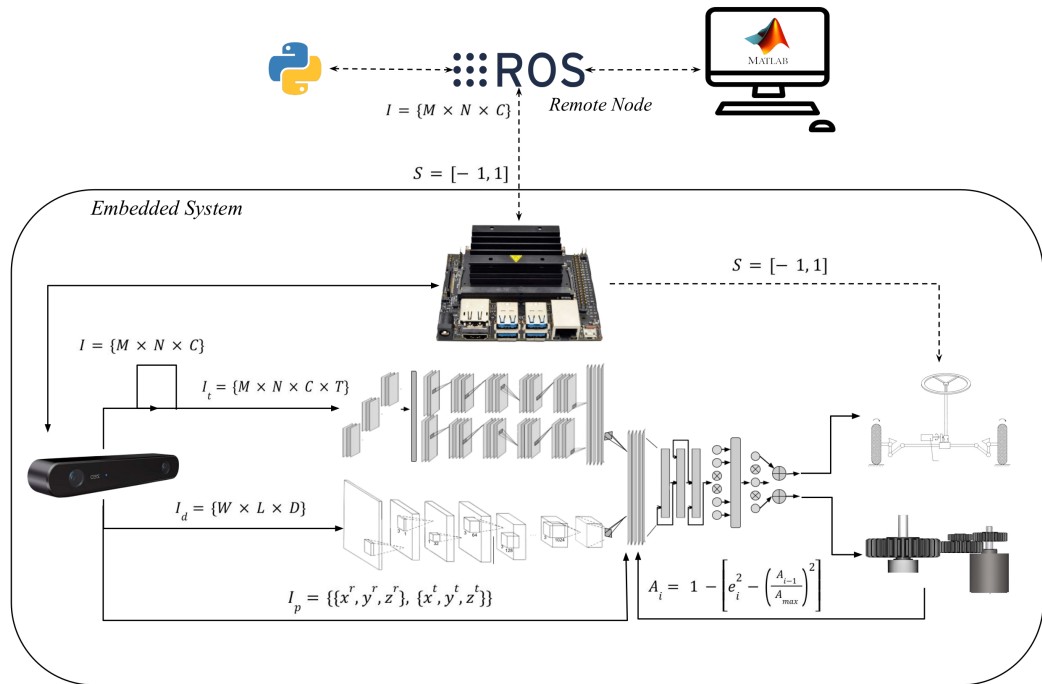

**Figure 10.** Embedded self-driving system operational diagram and remote interaction for comparison methods evaluation.

Due to the prototype characteristics, it was evaluated in simpler road scenarios than the simulation urban environments. The designed paths are inspired by the BlueRaven tracks for Jetbot and some obstacles were added to validate the ability to evade and correct the trajectory. Figure 11 details the designed roads in real-scale measurements.

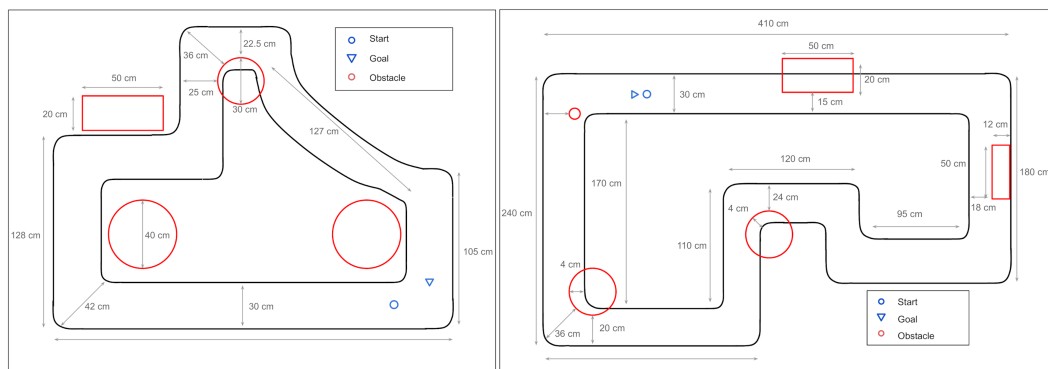

**Figure 11.** Real-scale track maps for physical self-driving tests.

In Figure 11, it can be seen that the paths have a width of 0.30 m, while in the simulation maps of Figure 7 these are exactly 4.8 m, so the tracks designed for this experiment are at 1:16 scale. In accordance with previously mentioned measurements of the prototype shown in Figure 9, both the test tracks and the vehicle are on the same scale, so the experimentation for the self-driving test it is considered viable in non-urban environment conditions at a scale of 1:16. In order for the displacement to be consistent with the track scale, the maximum speed was limited to 0.8 m/s, limiting the acceleration to:

$$A_i = 1 - \left[ e_i^2 - \left( \frac{A_{i-1}}{A_{max}} \right)^2 \right] \qquad (13)$$

where $A_{i-1}$ is the acceleration at the previous instant emitted by the neural model, on average at 50 Hz frequency. $e_i^2$ represents the rotation command of the current instant and

$A_{max}$ the maximum speed of 0.8 m/s, which in full scale would represent a maximum of 50 km/h. From the above, it is possible to observe the relationship between both experiments and their viability when evaluating the proposal and the comparison methods in the same way. Therefore, the results are discussed below.

Figure 12 shows the paths traveled on the test tracks. To increase experimentation, on the first track the obstacles location was modified to validate the evasion capacity. In the first scenario, all the methods were able to finish the course; however, when adding the obstacles, it was observed that Donkeycar did not overcome the second obstacle. On the more complex track both the Donkeycar and the Pilotnet model were unable to complete the full course, the latter evading all three obstacles but went off the road. In this case, MADT also goes beyond the path limits; however, it manages to resume autonomously. It can also be seen that both MADT and the proposed method generate very similar trajectories in the first scenarios. To make the comparison, a metrics summary is shown in Table 2.

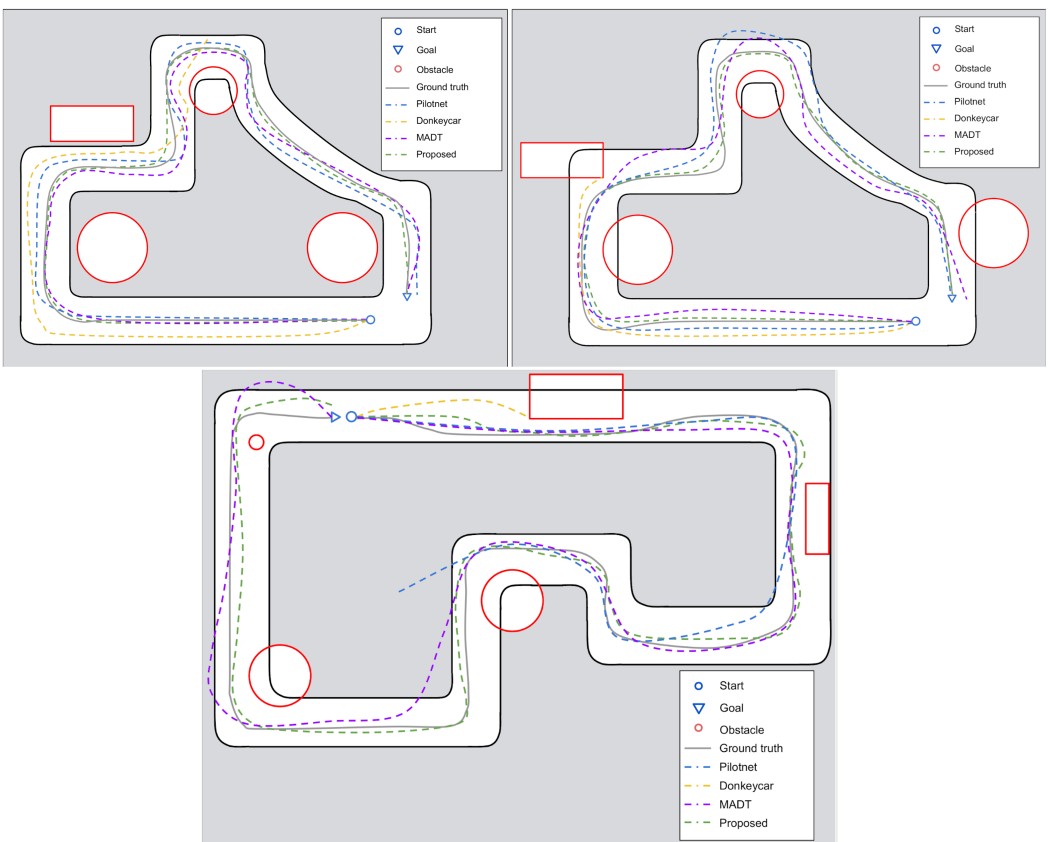

**Figure 12.** Trajectories obtained during self-driving in physical environments.

Table 2 shows the AAT of 75% for the Pilotnet model, this indicates that 25% of the path depends on human intervention, unlike the proposed method that depends only 5.6% if perfect driving is needed. Similarly, MADT receives an AAT of 89.8% as it went off track twice and needs to be corrected. In this experimentation, an autonomy of 89% was obtained, a decrease of 7.6% with respect to the autonomy in simulation is due to the fact that in the physical environment there are factors that alter perception such as the lighting changes, sensor noise, and affectations to control such as inertia and motors cumulative error. However, the proposed method is the least affected by this domain change, since the other methods decrease an average of 9.73% and up to 12.9% in the worst case. It can also be observed that the smoothness is reduced since a metric of 0.598 is obtained due to the path and physical effects complexity; however, the method is autonomous by 89%, which is better by $\approx$17.7% than the average of the other methods.

**Table 2.** Quantitative results of the experimentation on the scale prototype.

| Method | MSE | Cosine Distance | Behavior | Path Smoothness | Autonomy | AAT |
|---|---|---|---|---|---|---|
| Pilotnet | 0.185 | 0.171, −0.441 | 0.051 | 1.216 | 73.5% | 75.5% |
| Donkeycar | 0.205 | 0.322, −0.571 | 0.087 | 2.948 | 63.8% | 66.9% |
| MADT | 0.106 | 0.132, −0.100 | 0.030 | 0.834 | 78.3% | 89.8% |
| Proposed | 0.081 | 0.148, −0.075 | 0.027 | 0.598 | 89.0% | 94.4% |

Compared to human driving, the proposed method obtains a mean error of 0.081 and a smoothness of 0.589. For the aforementioned reasons, the metrics show higher performance and therefore lower performance. Even so, it is shown as the best by 0.025 with respect to the maximum and 0.084 with respect to the average, which is up to $2\times$ better than the rest of the comparison methods.

An important observation is that the proposed method can complete the path without interventions, compared to the Pilotnet model and the Donkeycar method, which require forced interventions to stay on track. From these experimental observations it can be inferred that the multiple data sources used improves the quality of self-driving, as well as that the neurofuzzy aggregation layer helps to decrease computational consumption making it possible to operate in a lightweight prototype. Furthermore, in this experimentation, the proposed method can be trusted by 89% in the worst case and 95% in a normal case. With these specific observations, the conclusions of this investigation can be reached.

## 4. Conclusions

Two main contributions are made in this work. The first is the formulation of a neurofuzzy aggregation layer for deep learning neural networks, which allows for composing new features from the several data sources fusion, reducing the computational cost while maintaining the precision of a multisensory system. The second contribution is the application of the proposed layer in the creation of a multisensory model for steering control. Additionally, the proposed model was tested in a simulated urban environments in ROS and in a scale prototype. The experimentation was carried out equally with our proposal and other methods of the state of the art. From this experimentation the following was observed: in simulation, an autonomy of 96.6% was obtained, which indicates that it depends on human intervention only 3.4% of the time. Regarding the smoothness of the road, a metric of 0.256 was obtained, demonstrating that the proposed method offers driving that is up to $2.8\times$ smoother and more comfortable. Regarding human driving, a difference of 0.021 is shown, which is 3.6 times more similar than the other methods. In the experimentation with the scale prototype, an autonomy of 89% was obtained. However, the proposed method is the one with the best performance, since the other methods only obtain an autonomy of 78.3% in the best case and 63.8% in the worst case. Compared with human driving, the proposed method obtains a mean error of 0.081 and a smoothness of 0.589, so it is shown to be the best by 0.025 with respect to the maximum and 0.084 with respect to the average, which is up to $2\times$ better.

In conclusion, the following advantages can be highlighted from this work: a model was created that performs data fusion and self-driving in a single process, based on the use of the proposed neurofuzzy aggregation approach, thus reducing the computational cost and allowing for its operation in real time. The proposed model is 95% reliable on average, with 2.8 better movement smoothness and 92% similarity to human behavior. The autonomous model completed the paths in 100% of the experiments. All this is because the two main contributions of this work offer the benefits of a multi-sensory system of several processes, but they are unified in a single one and reduce the cost to the point of being viable for use in real time in an embedded system. In future work, the model will be adapted to operate with more and different sensors for tasks such as pedestrians and traffic signs detection, as well as to integrate the capacity of planning and following routes. It is also intended to increase experimentation in different environments to demonstrate the feasibility of being implemented in a full-size prototype.

**Author Contributions:** Conceptualization, A.L.-Á., D.M.-V., M.M.-C., A.R.-C. and J.M.V.K.; methodology, A.L.-Á., D.M.-V., A.R.-C. and J.M.V.K.; software, A.L.-Á., M.M.-C. and A.R.-C.; validation, A.L.-Á., D.M.-V. and J.M.V.K.; formal analysis, A.L.-Á., D.M.-V. and J.M.V.K.; investigation, A.L.-Á., D.M.-V., M.M.-C. and J.M.V.K.; resources, A.L.-Á., D.M.-V., M.M.-C. and A.R.-C.; writing—original draft A.L.-Á. and D.M.-V.; writing—review and editing, A.L.-Á., A.R.-C., M.M.-C. and J.M.V.K.; visualization, A.L.-Á., A.R.-C. and J.M.V.K.; supervision, A.L.-Á., D.M.-V., M.M.-C. and J.M.V.K. All authors have read and agreed to the published version of the manuscript.

**Funding:** This research received no external funding.

**Institutional Review Board Statement:** Not applicable.

**Informed Consent Statement:** Not applicable.

**Data Availability Statement:** Not applicable.

**Acknowledgments:** The authors thank CONACYT, as well as Tecnológico Nacional de México/Centro Nacional de Investigación y Desarrollo Tecnológico for their support.

**Conflicts of Interest:** The authors declare no conflict of interest.

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
