# Peer review of "Neurofuzzy Data Aggregation in a Multisensory System for Self-Driving Car Steering"

_electronics, doi:10.3390/electronics12020314_

Round 1

Reviewer 1 Report

The article describes the Neurofuzzy Data Aggregation in a Multisensory System for Self-driving Car Steering. Although authors have covered all the aspects and explain them appropriately, the following modifications need to be considered for the possible publication of the manuscript. 

1. The abstract should reflect the background, significance, objectives, methodology, and conclusion briefly. Restructure the abstract accordingly. 

2. The article needs an extensive English editing and grammar check. Since, there are many missing links in between the Introduction section. 

3. The car (object) used in ROS simulation and protype are completely different in their dimensions. What about scalability and feasibility issues?

4. Highlight the novelty of the proposed work.

5. The conclusion should reflect the brief overview of results and their reasons. Restructure the conclusion section also. 

Author Response

A document with the comments addressed is attached.

Reviewer 2 Report

Various parts of the manuscript should be improved:

- Regarding name and idea of the manuscript: experimental tests were performed on mobile robot. Thus, in my opinion, name of the manuscript is misleading as it states "self-driving car".

- Abstract of the manuscript is too limited. The beginning of the abstract should be extended by in more general way explaining main idea of the manuscript.

- Self-driving car steering is very popular topic. Thus, the introduction / literature review is definitely too limited. Literature review should be performed in more detailed way.

- Major drawback – analysis / disccussion of the results. Simulations and experimental results are not properly disccussed. Manuscript lacks of proper disccussion regarding the results.

- Conclusions of the manuscript are too limited and do not properly represents the performed research. 

Author Response

(The authors gave the same response as above.)

Round 2

Reviewer 2 Report

No new comments. Good luck with Your future research.